# From Policy Reform to Public Reckoning: Exploring Shifts in the Reporting of Sexual-Violence-Against-Women Victimizations in the United States Between 1992 and 2021

**DOI:** 10.3390/bs15050701

**Published:** 2025-05-19

**Authors:** Jessica C. Fleming, Ashley K. Fansher, Ryan Randa

**Affiliations:** 1Department of Victim Studies, Sam Houston State University, Huntsville, TX 77341, USA; 2Department of Criminal Justice, University of North Dakota, Grand Forks, ND 58202, USA; ashley.fansher@und.edu; 3Department of Criminal Justice and Criminology, Sam Houston State University, Huntsville, TX 77341, USA; ryan.randa@shsu.edu

**Keywords:** sexual assault, reporting behaviors, violence against women, trends

## Abstract

The current literature indicates that sexual violence against women (VAW) is chronically under-reported to law enforcement due to factors such as fear of retaliation, societal stigma, and practical obstacles. Using National Crime Victimization Survey (NCVS) incident-level data, this study examines changes in the reporting patterns of sexual VAW from 1992 to 2021. This period of focus is notable for significant and, at times, unprecedented societal shifts and legislative reforms. Our results suggest that there are three distinct time periods for sexual VAW reporting in the United States, each marked by various social and political movements. These results provide researchers and law enforcement practitioners alike with insights into the instability of shifts in the reporting of sexual VAW to the police, supporting changes in how reporting behaviors should be viewed over varying time periods.

## 1. Introduction

Violence against women (VAW), as defined by the United Nations’ 1994 *Declaration on the Elimination of Violence Against Women*, includes gender-based violence causing physical, sexual, psychological, or economic harm in both public and private spheres, violating women’s autonomy and human rights ([115]). This definition builds on the earlier global recognition by the Committee on the Elimination of Discrimination against Women (CEDAW) in 1992 that VAW constitutes a form of gender-based discrimination, which laid the groundwork for framing VAW as a systemic and structural issue ([32]). Within the framework established by the 1992 CEDAW and the 1994 *Declaration on the Elimination of Violence Against Women*, sexual violence is a pervasive and uniquely damaging form of VAW, involving behaviors such as sexual assault, sexual harassment, and rape ([115]; [122]).

Despite the advancements in legislative reforms and governmental acknowledgement of these crimes, gaps remain in sexual violence research, particularly with regard to victims’ decisions to report violence and in understanding why victims often do not engage with the criminal justice system. For example, in the United States, 43.6% of women experience contact sexual violence,[note 1] and 21.3% report completed or attempted rape in their lifetimes, yet sexual violence remains chronically under-reported, obscuring its true societal impact ([107]). The exploration of victim decision-making has become an important area of focus within victimology, specifically what influences victim engagement with criminal justice agencies (e.g., [2]; [9]; [12]; [13]; [29]; [95], [96]; [107]).

Reporting incidents of sexual VAW to law enforcement, regardless of the reporter, initiates a formal criminal justice response. This step facilitates access to legal and support services for victims and ideally leads to holding perpetrators accountable ([25]; [36]). Furthermore, increased reporting of sexual VAW can challenge societal stigma and encourage other victims to come forward, fostering a culture of accountability and support ([9]; [46]). Although extant research has looked at the reporting of sexual VAW to police throughout the decades, what is lacking is a cohesive view of reporting of sexual VAW over time. In the nearly three decades that have followed the 1992 acknowledgment of VAW as gender-based discrimination ([32]; [115]; [122]; [124]), Americans have witnessed a rise in feminist activism, progressive legal reforms, and digital movements like #MeToo, which amplified public awareness of sexual violence, fostering cultural and policy shifts concerning VAW ([3]; [18]; [42]; [60]; [116]).

The present study employs thirty years of data from the United States-based National Crime Victimization Survey (NCVS) to examine trends and changes in the reporting of sexual VAW victimizations. The discussion will consider how these trends have been influenced by significant societal and legal events during the past 30 years.

### 1.1. Historical Context of Sexual VAW Reporting (Pre-1992)

Societal attitudes towards VAW over the past few decades have evolved, marked by growing awareness and condemnation of this behavior, along with transforming legislative policy, all influencing the reporting of VAW to the police. During the 1970s and 1980s, rape reform laws in the United States significantly altered the legal framework surrounding the treatment of rape victims, enhancing accessibility and fairness in the legal system. Key reforms included eliminating the corroboration requirement for victims’ testimonies, abolishing the prompt complaint rule,[note 2] and introducing “rape shield laws”,[note 3] which collectively helped reduce the stigma and barriers faced by victims. Research using National Crime Victimization Survey (NCVS) data and National Violence Against Women Survey (NVAWS) data shows that these legal changes increased reporting rates, demonstrating the importance of legislative reforms in encouraging victims to come forward ([9]; [21]; [29]).

Prior movements and the international recognition of VAW as gender-based discrimination in 1992 and a human rights violation in 1994 recognized the need for decisive legal action against VAW, setting a precedent for assertive federal action ([38]; [59]; [115]). Notably, the Violence Against Women Act (VAWA) of 1994 introduced comprehensive measures to address sexual violence in the United States ([17]; [53]; [102]). By recognizing both domestic violence and sexual violence as federal crimes and creating a comprehensive response system, the VAWA sought to reduce the stigma faced by victims and encourage a shift in societal and legal approaches toward VAW ([63]; [109]; [121]). Subsequent reauthorizations of the VAWA in 2000, 2005, 2013, and 2019 expanded protections and introduced measures targeting the needs of underserved populations ([52]; [69]). These updates also enhanced services for those experiencing VAW, reflecting a deepening understanding of VAW’s complexities and setting a precedent for federal support and protections ([17]; [52]; [63]; [84]). VAWA also represented a federal commitment to addressing VAW, boosting research funding, and sparking sustained academic interest in reporting challenges for victims and the criminal justice response.

From 1992 to 2021, several key Supreme Court decisions shaped the U.S. legal framework for addressing VAW. For instance, [117] ([117]) examined the federal government’s role in combating VAW, influencing the effectiveness of the VAWA and reshaping public and legal responses to VAW. Similarly, [33] ([33]) set clearer evidentiary standards, making it easier to use victims’ statements in court, which may have encouraged more victims to report incidents by offering better legal protections. These cases, among others, have collectively contributed to shaping how VAW is reported and addressed by law enforcement across the United States.

Alongside these legislative changes within the nearly three decades of the current study’s focus, media coverage of VAW varied, with ongoing calls by feminist activists, academic researchers, advocacy organizations, and public health organizations for more responsible reporting of VAW incidents and information ([41]; [77]; [110]).[note 4] Traditionally, VAW was reported as isolated and individualized incidents, but over the years, there has been a shift towards connecting incidents of VAW to broader societal issues ([41]; [57]). Indeed, the early 2000s and early 2010s demonstrated the media’s role in influencing societal attitudes towards VAW, victim-blaming, and, arguably, normalizing such violence, by using episodic framing, sensationalism, and misrepresentations that reinforced myths, while also blaming victims and relying on law enforcement as their primary source of information, thus normalizing VAW and hindering prevention efforts ([6]; [11]; [19]; [26]; [39]; [42]; [76]; [79]; [89]; [104]). Further, extant research has posited that increased public awareness and a broader definition of what constitutes rape, particularly regarding non-stranger and spousal incidents, has influenced and increased the reporting of rape from the 1970s until at least the end of 2000 ([12]).

Concurrently, the rise of digital activism has catalyzed significant changes in societal attitudes, legislative initiatives, and the reporting of VAW. [71] ([71]) quantify the impact of digital activism on reporting behaviors,[note 5] suggesting a marked increase in the formal reporting of sex crimes in the U.S. post #MeToo movement and emphasizing the tangible effects of digital platforms in fostering a conducive environment for reporting VAW. In Canada, a similar trend was observed, with 2017 witnessing a notable surge in sexual assault reports, peaking with the rise of the #MeToo movement ([101]).[note 6]

While the #MeToo movement of 2017 spurred public discourse related to the reporting of VAW, other digital feminist activism has also been influential. Launched in late September 2018 amid Dr. Christine Blasey Ford’s testimony alleging that a then-U.S. Supreme Court nominee sexually assaulted her in high school, the #WhyIDidntReport hashtag quickly captured public attention ([14]; [49]). The hashtag emerged in response to a tweet by the then-President, which implied that serious assaults should be promptly reported to the police, questioning the delay in Dr. Blasey Ford’s report ([14]; [49]). Within just two days, the #WhyIDidntReport hashtag saw approximately 675,000 tweets, with users disclosing their personal reasons for not reporting sexual violence to law enforcement ([14]). These tweets highlighted the complexity of victim-reporting decisions ([89]), supporting extant research indicating that the majority of sexual assault victims refrain from reporting to the police due to various factors ([10]; [74]; [114]; [123]).

### 1.2. Barriers to Reporting Sexual VAW

Despite the evolution that society and legislation has undergone since the 1970s, the landscape of sexual VAW reporting remains fraught with challenges. The current body of research demonstrates that reporting sexual VAW to the police has historically remained low ([9]; [12]; [13]; [29]; [85]). Victims often navigate a complex maze of barriers to reporting, including, but not limited to, fear of retaliation, societal judgment, and various practical obstacles, such as economic dependence on abusers, fear of financial instability after reporting, and high perceived legal costs, all highlighting the need for systemic reforms and trauma-informed approaches to support victims ([7]; [8]; [51]; [54]; [95]; [119]; [127]).

Moreover, sociodemographic factors such as race, ethnicity, education, and age intersect with these barriers, creating unique challenges for victims ([13]; [16]; [90]). For instance, minority women may encounter additional obstacles to reporting, such as language barriers and fears of deportation, particularly in contexts where immigration enforcement overlaps with law enforcement ([56]; [58]; [82]; [91]; [96]). Reporting decisions are also influenced by the nature of the crime, including factors like the degree of physical harm, weapon use, and the victim–offender relationship ([2]; [34]; [48]; [75]; [99]). Further, perceptions of prior interactions with law enforcement have been found to influence reporting decisions ([72]; [74]; [106]; [108]). Recent analyses of the #WhyIDidntReport movement also suggest that barriers to reporting may be dynamic, changing based on societal pressures and current political movements. For example, emotional and psychological factors, such as trauma, shame, and a lack of confidence in support systems, have recently become more prominent in discussions about why victims remain silent ([55]; [89]).

### 1.3. Trends in Reporting Patterns (1992–2021)

Prior research employing over 20 years of data has identified an increase in the reporting of VAW to the police ([12]; [13]; [29]). These studies document a rise in reporting and attribute the rise to various factors, including legislative changes aimed at improving the response to such violence ([12]; [13]; [29]). [12]’s ([12]) and [13]’s ([13]) investigations using NCVS data spanning from 1973 to 2005 found a significant increase in police notification by female rape victims, including victims of non-stranger rapes. [29] ([29]) further explored the effect of reforms in rape laws from 1974 to 1996, with an analysis of the National Violence Against Women (NVAW) Survey. Their exploration demonstrated an increase in rape reporting post-reform, indicating a positive impact of legal changes on reporting behaviors. However, the gap in reporting between simple and aggravated rapes during the study time frame remained, suggesting that the complexity of factors influencing victims’ decisions to report to police remains an area needing further exploration, particularly based on more recent available data.

### 1.4. Current Study

The current study quantitatively analyzes U.S.-based National Crime Victimization Survey (NCVS) data from 1992 to 2021, examining significant changes in sexual VAW reporting rates. Building on foundational studies, such as [12]’s ([12]) examination of rape reporting rates through 2000 and [29]’s ([29]) study on the impact of rape law reforms on reporting, this study extends the historical scope and trend analysis in sexual VAW reporting. Expanding the analysis up to 2021, this study will explore the potential changes in the propensity to formally report sexual victimizations against women, with the goal of providing a comprehensive view of sexual VAW reporting dynamics and patterns over time.

## 2. Materials and Methods

### 2.1. National Crime Victimization Survey

The U.S.-based National Crime Survey, first employed in 1972, was developed to collect data and provide estimates on a variety of crime victimizations that are not reported to law enforcement. In 1992, the survey underwent a major redesign of its methodology and became the National Crime Victimization Survey (NCVS). The NCVS has continued to undergo changes since 1992 to increase its sample size and the reliability of the data collected. These changes include transitioning to computer-assisted interviewing (2006) and tailored survey questions for improved data analysis (2006–2019; [24]).[note 7]

The NCVS collects data from a nationally representative sample of United States households, using multi-stage cluster sampling and a rotating panel design, adding more participants regularly. Approximately 150,000 households are currently in the sample, including an estimated 240,000 individuals aged 12 and older ([24]). Participants are interviewed every six months, for a three-year period, about their victimization experiences ([24]). In each interview, respondents are asked to report on incidents that happened in the last six months to prevent double reporting of experiences. Interview techniques and question wording are used to reduce issues with misreporting, double-counting, or other possible inaccuracies with participants’ recollection of events. A linked dataset, combining all data and reports from 1992 to 2021, was used for the current study.

The present analysis will only examine sexual victimizations of self-identified women, age 12 and older. The NCVS captures sex through a single variable. For the complete 30-year dataset, 114,496 male respondents and 161,885 female respondents were included. Males were removed from the data, resulting in only the self-identified female respondents remaining.

### 2.2. Variables

#### 2.2.1. Sexual Victimization

Within the NCVS, a single variable collects data on crime victimization incidents, encompassing violent and property crime, sexual and non-sexual. The variable was recoded following the guidelines[note 8] set by the 2016 *NCVS Technical Documentation* ([23]). The variable was recoded to only retain incidents of a sexual nature, including completed rape, attempted rape, sexual attack with serious assault, sexual attack with minor assault, sexual assault without injury, unwanted sexual contact without force,[note 9] verbal threat of rape, and verbal threat of sexual assault. After this recode, 2981 violent sexual incidents were retained for analysis. Sexual victimization represented approximately 10% of all VAW incidents in this dataset.

#### 2.2.2. Reporting

After a participant is acknowledged as experiencing a victimization event, they are asked, “Were the police informed or did they find out about this incident in any way?”, with options of “yes” (1), “no” (2), and “don’t know” (3).[note 10] Reporting in this sense could come from the person who experienced the crime or other third-party individuals. For the present analysis, only “yes” and “no” answers were used, removing 337 cases. Of the remaining 2483 sexual victimization incidents over thirty years of data, approximately 30% (n = 767) were reported to the police, and approximately 70% (n = 1716) were not.

#### 2.2.3. Year

Year was one of thirty values, ranging from 1992 to 2021, and was taken directly from the NCVS dataset. Table 1 displays the frequency and proportion of sexual victimizations by year.

### 2.3. Analytical Plan

The analytical plan for this study involves a multi-stage approach with three primary steps. Each of the first two steps is intended to elicit whether there is support for moving forward and continuing to pursue the following research question: Have the reporting patterns of sexual VAW victimizations to the police significantly changed from 1992 to 2021? The first step is to examine, through descriptive analysis in conjunction with chi-square tests, whether there is enough variation over the length of the study period to continue the analysis.

The second step includes the presupposition that there are clear and distinctive patterns in crime during the period from 1992 to 2021 that are associated with smaller blocks (i.e., eras) of time. For example, the well-known “crime drop” is a major trend-defined era within the time frame circa 1992–2006, and we also acknowledge that the end of the study period is marked by several important social movements that may manifest in the data as distinctive eras. To assess this possibility, we employ the Cochran–Armitage ([5]; [31]) test of trends with the goal of determining whether the data support isolating unique patterns.

Finally, should uniquely identifiable eras/trends emerge, we intend to quantify any observed differences, aggregated by era, through logistic regression and margins analysis, reported as the predicted probability. Overall, this approach not only allows us to identify patterns and shifts in the reporting of sexual violence against women, but also allows for the exploration of their alignment with societal and historical events, providing a comprehensive overview of reporting shifts across three decades.

## 3. Results

In the first stage, the chi-square analysis revealed significance in reporting variations across the years. For overall sexual VAW victimizations, Pearson (*χ*^2^(29) = 80.927, *p* < 0.001) and likelihood-ratio tests (*χ*^2^(29) = 79.258, *p* < 0.001) showed significant associations between the year and reporting status. These results indicate that there were substantial shifts in how frequently sexual VAW victimizations were reported to the police over the study period.

In the second stage of analysis, we apply LOWESS (Locally Weighted Scatterplot Smoothing), a technique that performs a locally weighted regression of each proportion variable on the annual year variable, creating smoothed values that help reveal underlying trends and patterns ([30]). Plotting the results allows users to visualize trends and identify inflection points, which, in this case, led to segmenting the total study period into three distinct eras: 1992 to 2001, 2002 to 2017, and 2018 to 2021. The sexual victimization reporting patterns each period can be seen in Table 2. This table displays the mean proportions of reported and unreported crimes.

After identifying these eras, we then conducted a series of OLS regression analyses to produce two different models and estimate their linear trends in the reporting behavior of victims of sexual-VAW. The predicted values from these regressions were then graphed alongside the observed proportions.

Next, we applied the Cochran–Armitage trend test to validate linear trends and detect deviations from linear patterns. For sexual victimizations, regression analyses showed a slight increase in the proportion of reported cases (*B* = 0.000, *p* < 0.001; *β* = 0.436) in addition to an increase in unreported cases (*B* = 0.001, *p* < 0.001; *β* = 0.499). This visualization is presented in Figure 1.

The results of the first two stages of the analysis are what we consider preliminary requirements for establishing the conditions necessary to evaluate whether any differences could be mapped onto a timeline that was constructed through the lens of social movements. Thus, in the final stage of analysis, we examined the reporting outcomes through logistic regression models to determine the possibility of differing effects across the three measured time periods. We report the predicted probabilities for ease of interpretation.

Our findings suggest significant shifts in reporting behaviors regarding sexual violence against women across the three distinct time periods. In line with prior research on the matter (e.g., [28]; [107]), victimization that is sexual in nature is more likely than acts of violence that are not sexual in nature to go unreported. As seen in Figure 1, at the height of victimization experiences in the United States (circa 1992), variability in reporting was at its greatest. As crime and victimization declined over the next decade, that variability decreased. An additionally noteworthy finding here is that, consistent with what we know about the ‘crime drop’ era, non-sexual instances of VAW were declining; however, sexual victimizations against women were not declining at the same rate, thus increasing their proportion among the total instances of VAW, while the gap between the proportions of reported and unreported victimizations was growing.

During the aggregated period of 1992–2001, the probability of reporting sexual victimization was 0.298. During the second aggregated period, 2002–2017, the probability of reporting sexual victimization increased to 0.362. During the final aggregated period, 2018–2022, a decline in the reporting of sexual victimization was revealed, resulting in a decrease in the likelihood of reporting to the police from 36% to 24% (−12.4%). These results can be seen in Table 3.

In sum, our results suggest that between 1992 and 2021, three distinct eras in reporting sexual VAW can be identified, and that at the calculated means, there are significant differences in the likelihood of reporting victimizations to the police. What is important to bear in mind is that the principle element of this calculus is the proportion of the total, rather than the actual, counts of victimizations, and as such, a somewhat more complex question arises that must address the trend in the volume of victimizations over time, and the interaction with variation in reporting.

## 4. Discussion

The current study explored the reporting patterns of sexual VAW to the police from 1992 to 2021 using a multi-step analytical approach. Our findings identified shifts in the reporting of sexual VAW patterns over the study period that supported the creation of three timeframes: 1992–2001, 2002–2017, and 2018–2021. The period from 1992 to 2021 in the United States is notable for numerous and, at times, unprecedented societal shifts and legislative reforms ([29]; [45]; [93]; [127]; [126]; [128]). While these events cannot be causally linked to changes in reporting patterns identified in the current study, they can provide context as to what was happening during the identified periods that may have influenced victims’ willingness and ability to report violence to the police.

The first period identified for the present study, from 1992 to 2001, also coincided with what has often been referred to as the ‘crime drop’ era due to substantial reductions in general crime rates across the United States ([44]; [70]). Concurrently, the identified time period is particularly notable in relation to VAW as it marks the culmination of feminist activism, which had been gaining momentum since the 1960s, leading to substantial shifts in legal protections and societal attitudes related to VAW ([15]; [20]; [83]). These broader societal shifts occurred alongside various factors contributing to the overall decline in crime rates, such as increased incarceration rates, improved policing strategies, the decline of the crack cocaine epidemic, and the legalization of abortion ([37]; [44]; [64]; [70]).

Coinciding with international efforts to combat VAW ([122]; [124]), this first time period also coincided with the enactment of the Violence Against Women Act (VAWA) in 1994, which introduced federal protections and resources aimed at increasing victim engagement with the criminal justice system ([17]; [63]; [83]; [102]). The second identified time period, 2002–2017, began just two years after the first reauthorization of the VAWA, where efforts continued through multiple reauthorizations, including a focus on addressing the needs of underserved populations, enhancing legal protections for victims, and mandating improved police practices, including specialized training and units for handling VAW cases ([102]; [105]). These changes, along with stronger collaboration with victim advocacy groups, may have contributed to the improved reporting of sexual VAW to the police identified within the second time period by assisting in creating a more supportive environment for reporting these crimes ([50]; [78]; [129])

This period also coincided with greater public discourse on sexual violence, policy reforms, and changes in institutional responses (e.g., Title IX expansions, reforms in victim services, and high-profile media coverage of sexual assault cases), along with heightened public awareness and activism surrounding sexual violence and early digital feminist movements ([18]; [42]). The influence and notoriety of public awareness campaigns (e.g., #MeToo) increased in intensity throughout the 2002–2017 time period, particularly in the latter years ([18]; [22]; [66]; [68]; [71]; [80]; [112]; [120]).

The current study’s findings of a slight uptick in reported sexual VAW from 2017 to 2018 (see Figure 1) align with the findings of [71] ([71]). Their analysis of U.S. Federal Bureau of Investigation National Incident-Based Reporting System (NIBRS) data from 2010 to 2018 found that the 2017 #MeToo movement led to increases in reported sex crimes, observing a 9% increase in reported sexual assaults during the initial 6 months of the movement, with effects sustained through December 2018 ([71]). While [71]’s ([71]) work provides important insight into the immediate impact of #MeToo on reporting behavior, which is reflected in the present study’s findings, it does not examine whether this trend persisted beyond 2018. Building upon these results and others, the current study’s analysis indicates that the upward trend observed by [71] ([71]) did not continue; instead, reporting rates dipped in the final identified 2018–2021 period.

The decline in the reporting of sexual VAW during the final time period identified in the present study (2018–2021) follows the peak of the #MeToo movement and coincides with the 2018–2022 lapse in the VAWA, the COVID-19 pandemic ([43]), and multiple highly publicized institutional failures in responding to sexual violence, among other societal movements ([27]; [47]; [55]; [81]; [89]; [103]; [118]). The 2018–2022 lapse of the VAWA ([103]; [111]) encompasses the entire span of the study’s final identified time period. It is important to note that funding for key VAWA programs continued during the lapse through congressional appropriations ([100]; [103]). However, the absence of formal reauthorization led to widespread advocacy efforts urging legislative action ([35]; [62]; [86]; [87]; [92]; [94]). The absence of VAWA reauthorization contributed to fiscal uncertainty about programing ([67]) and may have contributed to victim concern and confusion about available legal protections and support services, potentially influencing reporting behavior.

The 2018–2021 period also coincided with numerous high-profile media cases (e.g., Harvey Weinstein). The heightened media attention on cases of VAW during this time and the subsequent digital activism brought procedural justice issues to the forefront, which may have also contributed to the decline in reporting. For instance, digital activism movements during this period, like 2018’s #WhyIDidntReport, brought public attention to reasons why victims of VAW often avoid seeking help, emphasizing systemic and societal barriers and failures ([49]; [55]; [65]; [89]). Recent research comparing media portrayals during high-profile events, such as the Confirmation Hearings of Supreme Court Justices in 1991 and in 2018, illustrate a shift from prior time periods’ sensationalized, victim-blaming media narratives to depersonalized, politicized coverage ([1]).

A prevalent theme in the #WhyIDidntReport tweets was distrust of the justice system ([89]), likely fueled by the widely publicized treatment of Dr. Ford during the hearings, where her credibility was aggressively questioned and the proceedings seemed politically motivated ([88]; [89]). However, the decline in reporting noted in this later period compared to prior time periods may not necessarily indicate fewer incidents or decreased willingness to report, but rather, shifts in how individuals categorize their experiences and whether they view these incidents as warranting police involvement. Victims may now pursue alternative avenues for disclosure, such as social media or institutional reporting mechanisms (e.g., Title IX offices, workplace HR departments), rather than the criminal justice system.

### 4.1. Limitations and Future Research Directions

It is important to recognize that these findings should serve only as a starting point for understanding shifts in reporting sexual VAW to police. Further, while this study offers valuable insights, the findings are not without limitations. First, the present study employed a retrospective exploration of NCVS data, which, while nationally representative and capturing both reported and unreported victimizations ([40]), is limited as a self-reported survey. Relying solely on results from the NCVS may lead to a skewed understanding of VAW trends. Future research should consider corroborating NCVS data with multiple data sources like the Uniform Crime Reporting (UCR) program and the National Incident-Based Reporting System (NIBRS) to cross-validate findings and gain a more comprehensive understanding of shifts in sexual VAW reporting. By integrating UCR or NIBRS data, future research could assess whether increases in reporting resulted in more sexual VAW cases being documented, investigated, and resolved or if barriers at the institutional level continue to hinder case progression.

The decline in sexual VAW reporting from 2018 to 2021 coincides with multiple societal factors; however, the extent to which these factors influenced reporting decisions remains an open empirical question. Understanding VAW reporting requires considering how societal attitudes towards and awareness of VAW have evolved, which could be analyzed through the lens of media. It should also be noted that this period has the fewest data points in the analysis and should be interpreted with the appropriate cautions (see Table 3 for fluctuation in standard errors). Future research should consider analyzing media content alongside NCVS, UCR, and NIBRS data to provide a more comprehensive understanding of how media coverage and historical events influence sexual VAW reporting patterns, while also identifying gaps in the system where media-based interventions may improve justice outcomes.

This study examines changes in reporting patterns based on the proportion of victimizations reported to the police rather than the total number of victimizations. This means the findings capture differences in reporting likelihood but do not directly account for whether the overall number of victimizations increased or decreased over time. As a result, changes in reporting rates may be influenced not only by shifts in victims’ willingness or ability to report but also by fluctuations in the actual occurrence of victimization. Future research should examine how changes in the total number of victimizations over time may have influenced observed reporting trends.

Further, since this study aimed to identify patterns and shifts over time rather than fully explain the variance in reporting behaviors, key factors influencing sexual VAW reporting are not fully captured. For example, the current study is limited in its broad categorization of victimizations into sexual crimes, which overlooks the nuances in severity, a factor shown to influence reporting decisions ([2]; [10]; [12]; [46]; [48]; [75]). Indeed, severe sexual VAW incidents, such as those involving physical assaults, are more likely to be reported due to their tangible impact and physical evidence, whereas less severe incidents, such as verbal threats, often go unreported due to perceived insignificance or fear of disbelief ([2]; [4]; [13]).

Additionally, reporting behaviors are not uniform across all geographic areas, as structural and cultural factors can shape how sexual VAW cases are handled ([72]; [73]). While prior research has identified regional variations in VAW reporting ([97], [98]; [125]), the present study focuses on national trends and does not account for geographic differences. Future research should examine how regional disparities in culture, law enforcement practices, and resource availability influence reporting behaviors. Rural areas, for example, often lack specialized services for sexual assault victims, which may further discourage reporting ([73]). Analyzing geographic differences could help policymakers understand how localized policies and programs affect reporting, thus leading to more effective region-specific interventions.

### 4.2. Policy Recommendations

This study’s identification of three distinct periods can provide a basis for examining how institutional responses, policy changes, and societal attitudes may influence victims’ decisions to report. Although the present study does not explore or establish causal relationships for the shifts identified, its findings do identify distinct periods where the reporting of sexual VAW shifted. These results provide evidence that reporting has not moved in a continuous upward trajectory despite increased awareness of sexual violence and policy interventions. Policymakers and researchers alike should explore whether fluctuations in reporting patterns correspond with changes in media framing, public trust in institutions, and the availability of survivor-centered resources. Efforts to promote responsible media coverage of sexual violence, including discouraging victim-blaming narratives and amplifying institutional accountability, should be continued as a valuable component of broader policy initiatives aimed at fostering a culture that supports the reporting of sexual VAW.

However, since the findings of this study are based on the proportion of total victimizations reported rather than raw counts of victimization, they highlight a more complex issue regarding whether the decline in reporting was due to fewer victims coming forward, a true decline in victimization, or both. Practitioners and researchers alike should consider how these shifting patterns may reflect broader systemic issues and examine crime data alongside these results to determine which of these scenarios is occurring in order to adjust their strategies accordingly. For instance, if the proportion of reporting has indeed declined but victimization counts have stayed the same or increased, this may call for reassessing barriers to reporting (e.g., law enforcement policies, victim trust in the system, trauma-informed advocacy). On the other hand, if victimization counts themselves have declined, resources may be better allocated towards sustaining victim service programs rather than solely focusing on encouraging more reporting.

## 5. Conclusions

The present study expanded upon extant research (e.g., [12]; [13]; [29]) to provide a cohesive timeline of the reporting of sexual VAW to the police from 1992 to 2021. The results presented here identify that significant changes in the reporting of VAW to the police have occurred over the 30 years from 1992 to 2021, characterized by improvements and regressions in reporting within the most recent period. The patterns identified by the present study suggest that increased awareness and protections for victims of sexual VAW may not have necessarily translated into increased reporting of these victimizations in the United States. Shifts over time in reporting crimes to the police do not occur within a vacuum. Instead, they are intertwined with legislative changes, societal shifts, and the influence of social movements. Overall, these findings confirm and underscore the persistent under-reporting of sexual VAW and the need for ongoing research to bridge the gap between awareness and action in VAW reporting to the police.

## Figures and Tables

**Figure 1 behavsci-15-00701-f001:**
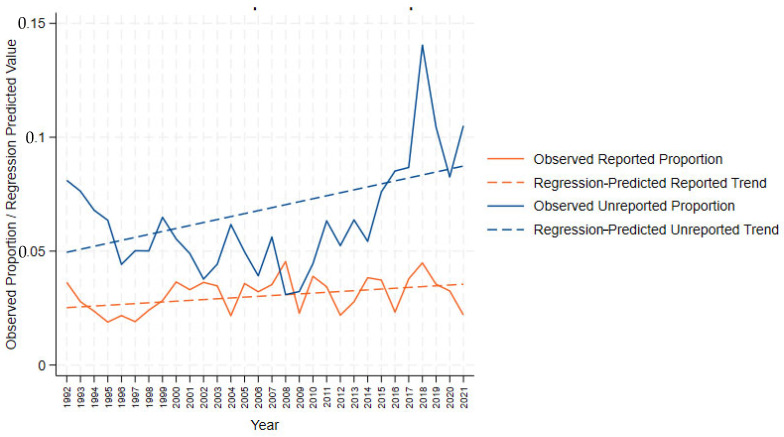
Reported vs. unreported sexual crimes (1992–2021).

**Table 1 behavsci-15-00701-t001:** Frequency and proportion of sexual crimes by year, 1992–2021.

Year	Sexual Crime Count	% Sexual Crimes Within Year	% of Sexual Crimes Across Dataset
1992	97	11.74	3.91
1993	131	10.41	5.28
1994	155	9.16	6.24
1995	118	8.24	4.75
1996	85	6.59	3.42
1997	80	6.92	3.22
1998	77	7.42	3.10
1999	89	9.31	3.58
2000	73	9.18	2.94
2001	62	8.20	2.50
2002	51	7.40	2.05
2003	50	7.90	2.01
2004	50	8.33	2.01
2005	43	8.55	1.73
2006	51	7.13	2.05
2007	57	9.15	2.30
2008	42	7.64	1.69
2009	29	5.50	1.17
2010	45	8.35	1.81
2011	54	9.76	2.17
2012	51	7.42	2.05
2013	56	9.15	2.26
2014	58	9.27	2.34
2015	79	11.33	3.18
2016	98	10.84	3.95
2017	135	12.45	5.44
2018	219	18.53	8.82
2019	146	13.98	5.88
2020	92	11.51	3.71
2021	110	12.70	4.43
Total	2483	9.69	100.00

**Table 2 behavsci-15-00701-t002:** Summary descriptive statistics by time period (n = 2483).

Reported Sexual Crimes	M	SD	Min	Max
1992–2001	0.026	0.006	0.019	0.036
2002–2017	0.033	0.007	0.022	0.045
2018–2021	0.035	0.008	0.022	0.045
Unreported Sexual Crimes				
1992–2001	0.061	0.012	0.045	0.081
2002–2017	0.057	0.018	0.031	0.087
2018–2021	0.111	0.021	0.031	0.140

**Table 3 behavsci-15-00701-t003:** Adjusted predictions of reporting sexual VAW crimes to the police by time period (n = 2483).

Time Period	Predicted Probability	SE	z-Value	95% CI [LL, UL]
1992–2001	0.298 ***	0.015	20.25	[0.269, 0.327]
2002–2017	0.362 ***	0.016	23.23	[0.332, 0.393]
2018–2021	0.238 ***	0.018	13.31	[0.203, 0.273]

Note. SE = standard error. *CI* = Confidence Interval. Predicted probabilities are based on margins analysis. The 95% confidence intervals indicate the range of expected true probabilities. *** *p* ≤ 0.001.

## Data Availability

The data are available in a publicly accessible repository. The data presented in this study are openly available online through the Inter-university Consortium for Political and Social Science Research (ICPSR) at https://doi.org/10.3886/ICPSR38430.v1 (accessed on 26 September 2023).

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
