# Peer review of "From Policy Reform to Public Reckoning: Exploring Shifts in the Reporting of Sexual-Violence-Against-Women Victimizations in the United States Between 1992 and 2021"

_behavsci, 2025, doi:10.3390/bs15050701_

Round 1

Reviewer 1 Report

Comments and Suggestions for Authors

Overview:

  • The current study utilized the National Crime Victimization Survey (NCSV), which collects data about crime victimizations from a nationally representative sample in the United States. The researchers used this dataset to analyze patterns of women’s reporting of sexual victimization to the police between 1992 and 2021C. They included a total of 2,483 sexual victimization incidents of which 30% were reported to the police. Their analyses revealed 3 periods of reporting: 1992-2001, 2002-2017, and 2018-2021. The probability of reporting was 0.298, 0.362, and 0.238 for the three periods showing an increase in reporting during the second time period and a decrease during the last period.

Strengths:

  • This is a well-written manuscript that contributes to the literature in interesting ways.

Limitations:

  • The aim appears to go beyond the methods and analyses but could be adjusted based on the research question that is narrower.

Introduction:

  • This is in general a well-written manuscript, including the introduction. One suggestion is to add a transition between the first and second paragraph as the end of paragraph one describes sexual violence as “a pervasive and uniquely damaging form of VAW” (p. 1) while the beginning of the second paragraph talk about advancements within the field. In the current version of the manuscript, it is difficult to follow this logic.
  • I wonder if the authors should consider revising their aim. It seems like the aim described in the introduction (on page 2, lines 61-63, and on page 5, lines 192-194) is broader than the actual analyses and results described in the methods and results sections. It seems like the research question on page 6, lines 247-248, better captures the aim of the study. The authors did analyze the patterns of sexual victimization between 1992 and 2021. In the discussion they provide some possible explanations for these patterns based on historical, legal, social, political, and cultural factors. However, if those explanations are part of the aim, then they also should be part of the method and result sections. In the current version of the manuscript, the authors do not provide a method to the contextual factors and how they determine which factors to focus on and why. This should be included if the authors want to keep the current aim.

Material and methods:

  • Please clarify what a linked dataset means (page 5, line 212).
  • The authors state that in the NCSV dataset participants are interviewed every 6 months for 3 years. I am assuming this means that one participant can report on the same or some other form of sexual victimization multiple times. How do the authors deal with participants’ report of multiple sexual victimizations?
  • The authors describe how the NCSV dataset has changed overtime. Do we know if these changes affect the results in any way? Both in terms of participants reporting sexual victimization experiences but also the analyses done in the current study.
  • Please clarify the age cutoff for those included in the current study. The authors mention that participants are 12+ in the NCSV dataset and that only self-identified women are included – are they 12 years and older?

Results:

  • Please clarify on page 8, lines 298-299, what the comparison group is (“…victimization that is sexual in nature is more likely to go unreported”).
  • The authors mention a few times in the manuscript, including in the result section, that there analyses of sexual victimization trends are related to the total number of victimizations reported to the police as opposed to the total number of victimizations. Why not do both? Or maybe control for the amount of victimizations?
  • The authors found three periods of reporting. The last period is shorter and only contains a few years – does this affect the results in any way? Is it possible that we “misinterpret” this last period due to the limited number of years included in the analysis?

Discussion:

  • I find the discussion well-written and includes several important points to consider both in trying to make sense of the results as well as important limitations to the current study (and possible alternative explanations).

Author Response

Reviewer 1:

  • The aim appears to go beyond the methods and analyses but could be adjusted based on the research question that is narrower.

 Introduction:

  • This is in general a well-written manuscript, including the introduction. One suggestion is to add a transition between the first and second paragraph as the end of paragraph one describes sexual violence as “a pervasive and uniquely damaging form of VAW” (p. 1) while the beginning of the second paragraph talk about advancements within the field. In the current version of the manuscript, it is difficult to follow this logic.
    • Thank you for this feedback. The first sentence of the second paragraph has been altered to better serve as a transition sentence between the two paragraphs.
  • I wonder if the authors should consider revising their aim. It seems like the aim described in the introduction (on page 2, lines 61-63, and on page 5, lines 192-194) is broader than the actual analyses and results described in the methods and results sections. It seems like the research question on page 6, lines 247-248, better captures the aim of the study. The authors did analyze the patterns of sexual victimization between 1992 and 2021. In the discussion they provide some possible explanations for these patterns based on historical, legal, social, political, and cultural factors. However, if those explanations are part of the aim, then they also should be part of the method and result sections. In the current version of the manuscript, the authors do not provide a method to the contextual factors and how they determine which factors to focus on and why. This should be included if the authors want to keep the current aim.
    • We agree with your comments here. On page 2, the last sentence of section 1.1 has been altered to remove any suggestion that the analysis examines cultural events. This language has also been removed from the Current Study section on page 5.

 Material and methods:

  • Please clarify what a linked dataset means (page 5, line 212).
    • The NCVS data has an individual dataset for each year. A linked dataset means that all the years were combined into one file. This clarification has been added.
  • The authors state that in the NCSV dataset participants are interviewed every 6 months for 3 years. I am assuming this means that one participant can report on the same or some other form of sexual victimization multiple times. How do the authors deal with participants’ report of multiple sexual victimizations?
    • The survey asks respondents to consider the last six months only, decreasing the likelihood of victimization events being reported multiple times. This language has been added to section 2.1.
  • The authors describe how the NCSV dataset has changed overtime. Do we know if these changes affect the results in any way? Both in terms of participants reporting sexual victimization experiences but also the analyses done in the current study.
    • The changes in 2006 and 2016 were the most significant. What is now footnote 6 elaborates on this may have altered the data reporting.
  • Please clarify the age cutoff for those included in the current study. The authors mention that participants are 12+ in the NCSV dataset and that only self-identified women are included – are they 12 years and older?
    • That is correct. This clarification has been added to the last paragraph of section 2.1.

 Results:

  • Please clarify on page 8, lines 298-299, what the comparison group is (“…victimization that is sexual in nature is more likely to go unreported”).
    • We have added that the comparison group will be VAW that is non-sexual in nature. 
  • The authors mention a few times in the manuscript, including in the result section, that their analyses of sexual victimization trends are related to the total number of victimizations reported to the police as opposed to the total number of victimizations. Why not do both? Or maybe control for the amount of victimizations?
    • This is a good observation by the reviewer. Of course, the main thrust of the work is to look at changes in the reporting trends themselves not necessarily any of the other elements that might spark interest when reading. I would also argue that by focusing on the proportion each year, rather than the volume, control for the total number of victimizations captured in the NCVS is part of the measures reported and discussed.
  • The authors found three periods of reporting. The last period is shorter and only contains a few years – does this affect the results in any way? Is it possible that we “misinterpret” this last period due to the limited number of years included in the analysis?
    • This is another good observation. I think it’s only fair to say that there is the possibility of a less reliable result for the noted time period and that does manifest in Table 2 where the readers will see that the standard error and Z values reflect that. Fewer data points can absolutely be worth noting and exercising caution when interpreting. We’ve added commentary to this effect in the limitations section of the manuscript.

 Discussion:

  • I find the discussion well-written and includes several important points to consider both in trying to make sense of the results as well as important limitations to the current study (and possible alternative explanations).
    • Thank you for this feedback!

Reviewer 2 Report

Comments and Suggestions for Authors

Thank you for the opportunity to review this excellent article. I have no hesitation in recommending that it be published in Behavioural Sciences. It discusses a very important topic in a scholarly way and offers important insights for the field. It is very well written, and clearly structured. The methodology is appropriate and well-explained. I like how the paper models that publicly available data are an excellent source for scholarly analysis. The article appropriately acknowledges its limitations and offers positive policy recommendations. As a result, it achieves its promise of expanding on extant research. The authors are to be commended.

Author Response

Thank you for the opportunity to review this excellent article. I have no hesitation in recommending that it be published in Behavioural Sciences. It discusses a very important topic in a scholarly way and offers important insights for the field. It is very well written, and clearly structured. The methodology is appropriate and well-explained. I like how the paper models that publicly available data are an excellent source for scholarly analysis. The article appropriately acknowledges its limitations and offers positive policy recommendations. As a result, it achieves its promise of expanding on extant research. The authors are to be commended.

  • Thank you for this review and support for our work.

Reviewer 3 Report

Comments and Suggestions for Authors

The article is a valuable empirical contribution to the field of victimology and the study of violence against women.

Among its strengths is the broad longitudinal approach (30 years) of the NCVS, an official source with standardized methodology.

Additionally, it provides an adequate historical contextualization of social movements (#MeToo, #WhyIDidntReport) and legislative changes (VAWA 1994).

However, it could be improved with a comparative table or graphic summary of the identified periods.

I congratulate the authors for their work

Author Response

  • The article is a valuable empirical contribution to the field of victimology and the study of violence against women. Among its strengths is the broad longitudinal approach (30 years) of the NCVS, an official source with standardized methodology. Additionally, it provides an adequate historical contextualization of social movements (#MeToo, #WhyIDidntReport) and legislative changes (VAWA 1994).
    • Thank you for this great feedback.
  • However, it could be improved with a comparative table or graphic summary of the identified periods.
    • A table has been added, now Table 2, showing reporting trends between time periods.
  • I congratulate the authors for their work

Round 2

Reviewer 1 Report

Comments and Suggestions for Authors

I appreciate the authors’ detailed response to the reviewers’ comments and suggestions. The revised manuscript addresses all of my concerns, and I would therefore recommend it for publication.